# The Therapeutic Effects of Lactic Acid Bacteria Isolated from Spotted Hyena on Dextran Sulfate Sodium-Induced Ulcerative Colitis in Mice

**DOI:** 10.3390/nu16213682

**Published:** 2024-10-29

**Authors:** Mengen Xu, Miao Hu, Jingbo Han, Lei Wang, Yuanyuan He, Md. F. Kulyar, Xiaohu Zhang, Yaozhong Lu, Siyang Mu, Hang Su, Jintao Cao, Jiakui Li

**Affiliations:** 1College of Veterinary Medicine, Huazhong Agricultural University, Wuhan 430070, China17386103281@163.com (M.H.); fakharealam786@hotmail.com (M.F.K.);; 2Department of Animal Nutrition and Feed Science, College of Animal Science and Technology, Huazhong Agricultural University, Wuhan 430070, China; 3College of Animals Husbandry and Veterinary Medicine, Tibet Agricultural and Animal Husbandry University, Linzhi 860000, China

**Keywords:** IBD, spotted hyena, LAB, colitis, tight junction, gut microbiota

## Abstract

Background: Inflammatory bowel disease (IBD) is a chronic condition characterized by recurrent episodes and an unclear etiology. Given the limitations of current therapeutic options, which include suboptimal efficacy and significant side effects, there is a pressing need to explore novel treatments. Probiotics derived from diverse species have been identified as a promising approach for managing IBD, owing to their anti-inflammatory properties and their ability to regulate gut flora, among other beneficial effects. Methods: In this study, three strains of lactic acid bacteria (LAB) were isolated from the feces of the scavenger spotted hyena (*Crocuta crocuta*), a scavenging mammal. Based on their capability to survive within and adhere to the gastrointestinal tract, along with their profile of antibiotic resistance, a high-quality strain of *Lactobacillus acidophilus* (LA) was selected and demonstrated to be safe for mice. Subsequently, the therapeutic efficacy of LA was evaluated using a dextran sulfate sodium (DSS)-induced model of ulcerative colitis in mice. Results: The results indicated that LA restored the disease activity index and improved histopathological lesions in the model group. It also reduced inflammation and oxidative stress and significantly restored the expression of mucins and intestinal tight junction (TJ) proteins (ZO-1, Occludin). Furthermore, LA corrected the DSS-induced disruption of the intestinal flora, leading to a significant decrease in the prevalence of potentially harmful bacterial genera, such as *Bacteroides*, and an increase in beneficial bacterial genera, including *Lactobacillus*. In conclusion, *Lactobacillus acidophilus* LA1, isolated from spotted hyena feces, has potential as a functional supplement for alleviating symptoms of IBD and regulating intestinal flora.

## 1. Introduction

Inflammatory bowel disease (IBD), which encompasses ulcerative colitis (UC) and Crohn’s disease (CD), is a chronic gastrointestinal disorder that is receiving increasing attention due to its rising prevalence and the lack of highly effective treatments [1]. Patients with IBD typically present with symptoms such as abdominal pain, bloody diarrhea, and fever. Diagnosis is confirmed through a combination of clinical evaluation, colonoscopic examination, and histopathological analysis [2]. The etiology of IBD remains complex and multifactorial, involving interplay between genetic predisposition, environmental factors, and alterations in the gut microbiota [3]. Current treatments for mild-to-moderate IBD include aminosalicylates, corticosteroids, and immunomodulators; however, these therapies often exhibit limited efficacy and are accompanied by significant adverse effects [4]. Given these limitations, there is a critical need for safer and more effective therapeutic approaches. Research into new strategies, such as the use of microbial agents, stem cell transplantation, and fecal microbiota transplantation, is ongoing. Among these, modulating the gut flora to alleviate IBD symptoms is emerging as a particularly promising area of investigation [5].

Probiotics are defined as live microorganisms that, when administered in adequate quantities, confer a health benefit on the host [6]. These microorganisms benefit host health through four principal mechanisms: (1) enhancement of intestinal mucosal barrier function, (2) prevention of pathogenic bacterial adhesion and colonization, (3) augmentation of the immune response, and (4) secretion of antimicrobial substances. Probiotics derived from yaks have been shown to enhance growth performance, antioxidant activity, and nonspecific immunity in mice [7]. The established efficacy of probiotics in repairing the intestinal barrier and restoring the balance of the gut microbiota highlights their potential for treating inflammatory diseases of the intestinal tract. *Lactic acid bacteria* (LAB) are particularly important probiotics due to their immunomodulatory effects, maintenance of intestinal microbial homeostasis, and growth-promoting activities, making them ideal candidates for further research [8]. The LAB isolated from Tibetan chickens enhanced growth performance, improved gut barrier health, and restored flora balance in broilers [9]. The study of LAB from various sources can yield more versatile and powerful strains, possibly offering new insights into the evolution and adaptation of these bacteria [10,11].

Hyenas, belonging to the order Carnivora, family Hyaenidae, are among the most common large carnivores in Africa and are listed on the IUCN Red List of Threatened Species. Four species of hyena are currently recognized: Spotted hyena (*Crocuta crocuta*), Striped hyena (*Hyaena hyaena*), Brown hyena (*Parahyaena brunnea*), and Aardwolf (*Proteles cristatus*) [12]. In addition to hunting fresh prey, hyenas are complex scavengers that primarily consume the carcasses of large vertebrates. Notably, the Aardwolf also feeds on termites [13]. The intricate and varied nature of their diet contributes to a similarly complex gut flora in hyenas. A study by Chen demonstrated that the fecal microbial diversity in spotted hyenas is influenced by factors such as sex, age, and feeding environment [14]. Moreover, spotted hyenas were found to have a higher abundance of the phylum Fusobacteria in their fecal microbiota compared to humans and other mammals. Comparative genomics of the striped hyena with nine other mammalian species revealed an expanded family of genes related to the immune and digestive systems, potentially representing an adaptive response to their scavenger lifestyle [12]. This suggests the presence of unique gut microbes in hyenas compared to other mammals. However, there is a scarcity of research on the gut microbes of hyenas, and even fewer studies have investigated the intestinal probiotics of these animals. In this study, three strains of lactic acid bacteria (LAB) were isolated from the fresh feces of adult spotted hyenas, and their stress tolerance and antibiotic sensitivity were assessed. Following the selection of a *Lactobacillus acidophilus* strain for safety evaluation, its role and mechanism in alleviating dextran sulfate sodium (DSS)-induced ulcerative colitis in a mouse model were investigated. This work aimed to identify novel probiotic strains that could mitigate the symptoms of inflammatory bowel disease (IBD).

## 2. Materials and Methods

### 2.1. Isolation and Identification of LAB from Hyenas

Fresh feces of adult spotted hyenas were collected from Wuhan Safari Park, Hubei Province, China, placed on ice, and transported to Huazhong Agricultural University, Wuhan, China. Thereafter, 2 g of feces were mixed thoroughly with sterile phosphate-buffered saline PBS (Biosharp, Beijing, China) and incubated in a bacterial shaker at 200 rpm for 30 min at 37 °C. The supernatants of the mixture after settling were streaked on De Man, Rogosa, and Sharpe (MRS) agar plates (Beijing Land Bridge Technology Company Ltd., Beijing, China) and incubated in a biochemical incubator at 37 °C for 48 h. Single colonies of suspected LAB, exhibiting a milky-white and raised appearance, were selected and purified for three consecutive generations. After that, single colonies were picked and inoculated in MRS broth for 24 h. All isolates were identified through a combination of morphological characterization and Gram staining. Genomic DNA was extracted from the isolates using a bacterial DNA extraction kit (Foregene, Chengdu, China). The 16S rRNA gene of the isolates was then amplified using universal primers 27F and 1492R [15].

### 2.2. Growth Curve Mapping of Isolates

Isolates were inoculated in MRS broth medium and incubated in a shaker at 37 °C for 24 h at 200 rpm. During this period, the absorbance was measured at 600 nm (OD600) at the beginning of the day at 2 h intervals, with three repetitions of each measurement.

### 2.3. Tolerance of Isolates to Artificial Gastric Intestinal Fluids

The isolates that were inoculated in MRS broth medium and incubated in a shaker at 37 °C for 12 h at 200 rpm were inoculated in artificial gastric and artificial intestinal fluids (Yuanye, Shanghai, China) at a concentration of 1% (*v*/*v*) and incubated in MRS broth medium as control at 37 °C for 3 h, and then bacterial viability was calculated using the plate counting method according to the following formula:$$Survival\;rate\;\left(\%\right)=\frac{No.\;of\;colonies\;treated\;with\;gastrointestinal\;fluid}{No.\;of\;colonies\;in\;the\;control\;group\;}\times 100$$

### 2.4. Hydrophobicity and Autoagglutination of Isolates

#### 2.4.1. Hydrophobicity

The isolates were inoculated in MRS broth and incubated at 200 rpm for 12 h in a shaker at 37 °C. After that, centrifugation was performed at 8000 rpm for 10 min, forming a bacterial precipitate. Bacterial suspensions were prepared by resuspending the organisms following two washes with sterile PBS. The OD600 of the bacterial suspension was recorded as *A*0. The bacterial suspension was combined with an equal volume of xylene and incubated at 37 °C for 30 min. The aqueous phase was then carefully aspirated, and its OD60 was determined and recorded as *A*1. The measurement was repeated three times. The following formula was used to calculate the hydrophobicity of the isolates:$$Hydrophobocity\;\left(\%\right)=\left[1-\left(\frac{A1}{A0}\right)\right]\times 100$$

#### 2.4.2. Autoagglutination

Following the preparation of the bacterial suspension as per the methodology above, it was incubated at 37 °C for 2 h. The OD600 of the supernatant was measured and recorded as *B*1. Subsequently, the autoagglutination of the isolate was calculated according to the following equation, which was derived from the results of three repeated measurements:$$Autoagglutination\;\left(\%\right)=\left[1-\left(\frac{B1}{A0}\right)\right]\times 100$$

### 2.5. Antibiotic Susceptibility of Isolates

100 μL of the culture, which had reached the plateau stage, was evenly distributed on MRS agar plates until the culture was fully absorbed. Following the antibiotic disc’s application to the plates’ reverse side, the latter were inverted and incubated at 37 °C for 24 h. After that, the diameters of the inhibition circles were measured.

### 2.6. Safety Experiments with Isolated LA

The safety of a final screened strain of *Lactobacillus acidophilus* LA was evaluated using mice as experimental animals. Twenty-four 3-week-old C57BL/6 mice, acclimatized for one week, were randomly and evenly divided into a control (C) group and a *Lactobacillus acidophilus* (LA) group, with 12 mice each. The LA group was administered LA via gavage at a dose of 1 × 10⁹ CFU/d for 14 consecutive days, while the control group was gavaged with an equal volume of sterile MRS broth. The mice were observed and recorded daily throughout the experiment. This entailed monitoring their vitality, behavior, coat condition, constipation, food intake, and body weight. After 14 days, all mice were euthanized, and a series of biological specimens were collected for analysis. These included blood, liver, spleen, and jejunum samples.

### 2.7. Animal Experiments

Thirty-six 6-week-old C57BL/6 mice were randomly assigned to three groups following one week of acclimatization. The groups were the no treatment (NT) group, DSS treatment (DSS) group, and DSS+LA treatment (DSS-LA) group. IBD was induced in mice by adding 2.5% DSS to the drinking water of the DSS and DSS-LA groups between days 0 and 7. In contrast, no DSS was added to the drinking water of the NT group. During the period between days 8 and 14 of the study, the DSS-LA group was administered LA via gavage at a dose of 1 × 10⁹ CFU/d/each, while the NT and DSS groups were gavaged with equal amounts of sterile MRS broth. Throughout the experiment, the body weight, health status (including coat condition, presence of constipation, blood in stool, mental status, etc.), and mortality of the mice were observed and recorded. The severity of colitis was evaluated by calculating the disease activity index (DAI) in mice based on the following criteria: weight loss, constipation, and the presence of blood in the stool [16]. The mice were euthanized by cervical dislocation on day 14, and the serum was obtained by centrifugation following the collection of mouse blood and stored at −80 °C. The length and weight of the colon were measured and subsequently divided into two parts: the proximal colon was stored at −80 °C, and the distal colon was fixed in paraformaldehyde.

### 2.8. Histological Staining and Immunohistochemistry

The colon tissue was fixed in paraformaldehyde for 48 h, after which it was cut into slices of 0.5 μm after ethanol dehydration and paraffin embedding. The sections were stained with hematoxylin–eosin and Alcian blue/periodic acid–Schiff to assess the morphological changes in the colonic tissue and the alterations in the number of goblet cells [17]. Furthermore, immunohistochemical staining of tissue sections was employed to determine the expression of colonic tight junction proteins, namely ZO-1 and Occludin [18].

### 2.9. Measurement of Serum Levels of Inflammatory Factors and Oxidative Stress

To evaluate the extent of systemic inflammation and oxidative stress in the mice, the serum levels of inflammatory cytokines including tumor necrosis factor α(TNF-α), Interleukin-1β (L-1β), Interleukin-6 (IL-6), antioxidant markers superoxide dismutase (SOD) and oxidative stress marker malondialdehyde (MDA) were quantified using the methodology outlined by the reagent vendor [9] (Jiancheng, Nanjing, China).

### 2.10. Gene Expression Analysis

According to the methodology outlined by Liu et al. [19], 2 g of RNA was extracted from colon tissue using Trizol reagent, and cDNA was generated through reverse transcription with Hiscript III Reverse Transcriptase (Vazyme, Nanjing, China). Quantitative PCR was conducted using an AceQ Universal SYBR qPCR Master Mix (Vazyme, Nanjing, China), and the resulting data were analyzed on a LightCycler 96 apparatus (Roche, Basel, Switzerland). The relative changes in target genes were calculated using the 2^−ΔΔCt^ method, with GAPDH as the internal reference gene. The primers used were synthesized by Sangon Biotech Co. Ltd. (Shanghai, China). and are listed in Table 1.

### 2.11. 16S rRNA Sequencing Analysis

The total DNA was extracted from the collected mouse cecum contents using the QIAamp DNA Mini Kit (Qiagen, Dusseldorf, Germany), and the concentration and purity of the extracted DNA were then assayed using a UV spectrophotometer (Thermo Scientific, MA, USA). A pair of primers were synthesized based on the V3/V4 region of 16S rRNA (338F: ACT CCT ACG GGA GGC AGC A and 806R: GGA CTA CHV GGG TWT CTA AT) [20]. The PCR products were purified and recovered using Vazyme VAHTSTM DNA Clean Beads and subsequently analyzed with quantitative PCR using a microplate reader (FLx800, BioTek, Winooski, VT, USA). The samples were combined in the optimal proportions based on the qPCR outcomes and the requisite sequencing volume for each sample. Subsequently, the Illumina TruSeq Nano DNA LT Library Prep Kit (Illumina, San Diego, CA, USA) was employed to prepare sequencing libraries, and high-throughput sequencing was conducted on the QC-qualified libraries.

### 2.12. Statistical Analysis

All data in this study were statistically analyzed and plotted using GraphPad Prism 8.0 software. Data are expressed as the mean ± standard deviation (SD), and multiple group comparisons were made using ordinary one-way analysis of variance (ANOVA). A *p*-value of less than 0.05 was considered statistically significant.

## 3. Result

### 3.1. Isolation, Identification, and Growth Characteristics of Hyena-Derived LAB

Following incubation, neatly edged, translucent, milky-white colonies of suspected LAB were selected from MRS agar plates. Three Gram-positive strains were then screened and analyzed in conjunction with the staining results (Figure 1A). The 16S rRNA sequences of the three strains were amplified and detected using 1% agarose gel electrophoresis. The amplified products were found to be as expected (Figure 1B). The amplification products were subjected to homology analysis. A phylogenetic tree was constructed (Figure 1C). A combination of morphological characteristics and 16s rRNA results identified three strains as *Lactobacillus salivarius* (LS), *Lactobacillus reuteri* (LR), and *Lactobacillus acidophilus* (LA). All three strains cultured in MRS broth exhibited exponential growth from 2 h onwards, reaching a plateau at 10 h before stabilizing (Figure 1D).

### 3.2. Survival and Colonization of Isolates in the Gastrointestinal Tract

The capacity of the isolates to survive within the gastrointestinal tract was evaluated by measuring their survival in artificial gastric and intestinal fluid [21]. The results demonstrated that LR exhibited the most significant resistance to the simulated gastrointestinal environment, with survival rates of 87.18 ± 7.29% and 68.24 ± 5.43% in artificial gastric and intestinal fluids, respectively. The second highest rate was observed in LA, with 26.31 ± 1.19% and 30.06 ± 2.37%, respectively. The lowest survival rate was observed in LS, with values of 55.44 ± 5.31% and 14.89 ± 4.25%, respectively (Figure 2A,B). The properties of hydrophobicity and autoagglutination can partially indicate the capacity for bacterial adhesion. LS exhibited the highest hydrophobicity, at 87.58% ± 1.74%. LR followed this, demonstrating a hydrophobicity of 87.09% ± 3.05%. In contrast, LA exhibited the lowest hydrophobicity, with a value of 63.73% ± 4.05%. Regarding autoagglutination, the highest level of LA was observed at 85.49% ± 1.82%, followed by LR and LS at 63.75% ± 0.93% and 63.05% ± 1.59%, respectively (Figure 2C,D).

### 3.3. Findings of Antibiotic Susceptibility Tests

The results demonstrated that LA exhibited sensitivity to all antibiotics subjected to testing, whereas LR displayed resistance to a subset of antibiotics, and LS demonstrated resistance to all antibiotics tested (Table 2).

### 3.4. Safety Assessment of Isolated LA

Throughout the experimental period, the mice in both groups exhibited optimal mental health, standard behavioral patterns, and balanced daily food and water intake. They did not experience any instances of diarrhea or mortality. Furthermore, no notable discrepancies between the control and probiotic groups were observed in body weight and liver and spleen indices (Figure 3A–C). HE staining of the jejunum and liver revealed no evidence of significant inflammation or damage to the liver or intestine following the administration of LA (Figure 3D,E). Additionally, there were no significant differences in villus length and crypt depth (Figure 3F). The results show that LA is a safe compound in vivo in murine models.

### 3.5. Effect of Isolating LA on the Physiological Indices in Mice

Following the administration of DSS on day 5, a reduction in body weight was observed in mice in the DSS and DSS-LA groups, whereas the weight of mice in the control group remained stable. In contrast, gavage LA mitigated the weight loss associated with DSS and promoted weight regain (Figure 4A), and the increase in DAI resulting from DSS administration was also significantly reduced by gavage LA (Figure 4B). A notable consequence of colitis induced by DSS is the shortening of the colon [22]. The administration of Gavage LA resulted in a noteworthy improvement in colon length compared to the DSS group (Figure 4C,D). Hepatomegaly and splenomegaly may be observed in response to colitis [23]. The administration of LA treatment has been demonstrated to reduce elevated liver and spleen indices caused by colitis (Figure 4E,F). The results above illustrate that LA can mitigate the clinical manifestations and severity of colitis in mice induced by DSS.

### 3.6. Effect of Isolating LA on Colon Histology

The impact of LA on colonic morphology was evaluated through HE staining. The results demonstrated that DSS induced acute inflammation and histological damage to the colon, as evidenced by loss of colonic crypts, substantial inflammatory cell infiltration, and shortening or loss of intestinal villi. In contrast, the degree of damage to the villi and crypts, as well as the infiltration of inflammatory cells, was less severe in the DSS-LA group than in the DSS group (Figure 5A). The identification of the quantitation of mucin secreted by them in the colon, which is indispensable for the intestinal defense mechanism, was conducted through the use of AB-PAS staining. The AB-PAS sections demonstrated that DSS reduced the quantity of mucin present within the colon, with a notable increase following the administration of LA (Figure 5B). The intestinal epithelium preserves its barrier integrity primarily through TJ proteins, mainly ZO-1 and Occludin [23]. The immunohistochemical analysis revealed significantly lower expression of the two TJ proteins (*p* < 0.05) in the DSS group compared to the control group. Conversely, the DSS-LA group exhibited significantly higher expression of these proteins (*p* < 0.05) than the DSS group (Figure 5C). The same outcomes were evident in the expression of mRNA (Figure 5D). Overall, LA reverses the histological damage to the colon as well as the loss of mucin and TJ proteins caused by DSS, thereby providing a therapeutic effect on colitis.

### 3.7. Effect of Isolated LA on the Serum Levels of Inflammatory Factors and Oxidative Stress in Mice

The determination of inflammation in the mouse organism involves examining the serum levels of inflammatory factors [9]. The serum biochemical results demonstrated that DSS supplementation elevated the serum levels of three inflammatory factors, TNF-α, IL-1β, and IL-6, in the mice (*p* < 0.01). Conversely, the serum levels of those above three inflammatory factors were markedly diminished following the gavage of LA (*p* < 0.01) (Figure 6A–C). A significant elevation in MDA was observed in the DSS group compared to the control group (*p* < 0.01). Conversely, LA demonstrated the ability to reduce the level of MDA (*p* < 0.01). However, no notable discrepancy was identified in the level of SOD between the DSS and DSS-LA groups (Figure 6D,E).

### 3.8. Effect of Isolated LA on the Intestinal Flora of Mice with DSS-Induced Colitis

Analysis of intestinal flora composition was performed via 16S rRNA gene sequencing to assess the impact of LA on the improvement of intestinal flora structure. Wayne plots showed that the number of OUTs shared by the three groups was 660, while the number of OUTS exclusive to the control, DSS, and DSS-LA groups was 5532, 3657, and 4108, respectively (Figure 7A). α-diversity is responsive to species richness and diversity, while the Chao1 index, Simpson index, and Shannon index are measures of microbial diversity. The NT group exhibited significantly higher microbial diversity than the DSS group (*p* < 0.01). In contrast, the DSS group demonstrated significantly lower microbial diversity than the control group. Unfortunately, there was no statistically significant difference between the DSS and DSS-LA groups (Figure 7B). The β-diversity results demonstrated that, based on principal coordinate analysis (PCoA), individuals in each group were clustered together, with the DSS group exhibiting a clear separation from the NT group. This finding substantiates the hypothesis that DSS affects the composition of the gut flora. Conversely, individuals in the DSS-LA group exhibited closer proximity to the NT group, indicating that LA effectively restored the DSS-induced disruption of the gut flora (Figure 7C).

The specific composition of microbial communities at different taxonomic levels is of paramount importance for the evaluation of the function of probiotics [24]. The abundance of the top 20 gut microbial communities was analyzed between groups at the phylum and genus levels. At the phylum level, *Firmicutes*, *Bacteroidota*, and *Verrucomicrobiota* were the most prevalent phyla across all three groups. The proportion of Proteobacteria containing multiple pathogens was significantly higher in the DSS group than in the control group. Conversely, the proportion of Proteobacteria was observed to be significantly lower in the DSS-LA group (NT = 0.75%, DSS = 14.02%, DSS-LA = 4.88%) (Figure 7D). At the genus level, the DSS group of *Bacteroides*, *Lachnoclostridium*, *[Clostridium]_innocuum_group* exhibited a notable increase in abundance, whereas *Lactobacillus*, *Candidatus_Saccharimonas*, *Odoribacter* demonstrated a significant decrease in abundance, which was corrected in the DSS-LA group (Figure 7E,F) (*p* < 0.05 and *p* < 0.01). Similar results were found in the Lefse analysis and LDA analysis (LDA > 3) (Figure 7G). These results indicate that the abundance of health-threatening bacteria in the gut decreases while the abundance of beneficial bacteria increases after LA supplementation.

## 4. Discussion

As a disease with increasing incidence, the causes of IBD are recognized as environmental, genetic, gut microbiota dysbiosis, and immunomodulatory factors [1]. Current treatments for IBD are primarily hormones, antibiotics, immunosuppressants, and the removal of inflamed intestinal tissue. However, these traditional treatments have proven ineffective and prone to adverse and destructive effects [25]. Consequently, there are growing efforts to identify new treatments that are safe and effective. Liu et al. [19] demonstrated that tea extracts exhibited anti-inflammatory and protective effects against colitis and liver damage in a DSS-induced colitis model. These effects were attributed to restoring intestinal barrier integrity and modulation of intestinal flora disorders. Furthermore, the unfermented tea extracts had superior therapeutic efficacy compared to their fermented counterpart. The findings of Zhong et al. [26] highlight the pronounced efficacy of *Sanghuangporus* in alleviating DSS-induced colitis by promoting *A. onderdonkii* and 5HIAA, which ultimately activates AhR signaling. In addition to the previously discussed plant extracts, probiotics derived from various species have been demonstrated to have beneficial therapeutic effects in IBD. Qin et al. [27] isolated and screened a strain of *Lactobacillus plantarum* A3 from the intestinal tract of horses that can alleviate typical colitis symptoms, including a shortened colon and reduced DAI index in mice caused by DSS. These findings offer promising insights into the potential treatment of inflammatory bowel disease (IBD). The *Lactobacillus plantarum* HY7718 strain, isolated from fermented squid, has been demonstrated to alleviate symptoms of colitis in mice through regulation of the TLR-MyD88-NF-κB signaling pathway and the promotion of anti-inflammatory cytokine expression [16]. In this study, mice administered DSS showed weight loss with signs of blood in their stools, shortening of the colon, enlargement of the liver and spleen, and an elevated DAI index, consistent with the characteristics of IBD. After LA supplementation, the mice showed significantly better body weight reduction and bloody stool symptoms, a significantly lower DAI index compared to the DSS group, and a significant increase in colon length, demonstrating the feasibility of LA as a therapeutic agent for IBD.

The spotted hyena exhibits a sophisticated intestinal flora composition and is a significant scavenger. The age, sex, and husbandry environment influence the intestinal flora of the spotted hyena [14,28]. The spotted hyena is predominantly distributed across the African continent. Most Chinese spotted hyenas are bred in captivity, exhibiting a notable divergence in the composition of their intestinal flora compared to their wild counterparts. Nevertheless, research on captive spotted and wild hyenas’ intestinal probiotics is limited. Given the distinctive characteristics of the species and its diet, it is hypothesized that the intestinal probiotics isolated from spotted hyenas will exhibit unique functions. In the present study, three strains of LAB were isolated from fresh feces of adult spotted hyenas and identified through a combination of microscopic examination, Gram staining, and 16S rRNA analysis.

Probiotics must resist compression by gastric and intestinal juices to survive and adhere to the intestinal tract, where they can function [29]. The results showed that isolated LR exhibited the most significant resistance to the simulated gastrointestinal environment, with survival rates approaching 90% in gastric juice and 70% in the intestine. In addition to resisting the pressure exerted by the gastrointestinal tract, probiotics must adhere to the intestinal tract to be effective. The adhesive capacity of probiotics is typically assessed in terms of their hydrophobicity and self-agglutinating ability [30]. Prior research has indicated that strains exhibiting high hydrophobicity and self-agglutinating capabilities demonstrate enhanced capacity to adhere to the gastrointestinal tract. Probiotic strains alter the bacterial composition of a polar environment through hydrophobicity and subsequently act through autoagglutination by aggregating with each other to a certain amount, forming a barrier that prevents the colonization of pathogenic bacteria [31]. All three strains in this study had excellent hydrophobicity and self-agglutination ability, with LS having the highest hydrophobicity and LA having the highest self-agglutination ability. Due to their efficacy in combating disease and stimulating animal growth, antibiotics have been extensively employed in the agricultural sector. In addition to the substantial economic advantages, the inappropriate use of antibiotics has resulted in many intractable issues, including the emergence of antibiotic-resistant bacteria and the transfer of antibiotic-resistant genes [32]. A report by O’Neill indicates that 700,000 deaths occur annually as a result of infections with drug-resistant pathogens. This figure could reach 10 million by 2050 if action is not taken to address the issue [33]. Probiotics are increasingly being employed as an alternative to antibiotics. However, some recently identified probiotics have also developed partial antibiotic resistance due to the selective impact of pre-existing antibiotic usage on probiotics within the gut [34]. To prevent the transfer of resistance genes to other pathogenic bacteria and their acquisition of resistance, the European Food Safety Authority (EFSA) recommends that probiotics should not contain acquired or transferable resistance genes encoding common antibiotics in clinical use [35,36]. Consequently, the issue of antibiotic resistance has become a significant element in the evaluation of the safety of probiotics. Of the three strains isolated in this study, LA demonstrated susceptibility to all common antibiotics tested, whereas the other two strains exhibited resistance to a wide range of clinically used antibiotics. This finding was a key factor in selecting LA for subsequent experiments. Subsequently, the safety of LA was determined through in vivo experimentation. During the experiments, no abnormalities were observed in the mice, including any morbidity or mortality and visceral or intestinal lesions. Furthermore, the experimental and control groups had no significant difference in body weight.

The intestinal mucosal barrier, which encompasses mechanical, chemical, immune, and biological barriers, represents the initial interface between the gut and the external environment [37]. TJ proteins play a crucial role in maintaining the integrity of the mechanical barrier by sealing the intercellular space between neighboring intestinal epithelial cells [38,39]. This prevents the intrusion of bacteria or pathogens into the intestinal mucosal lamina propria, which could otherwise lead to an aberrant mucosal immune response. Excess inflammatory factors and reactive oxygen species destroy TJ proteins, thereby facilitating the erosion of the intestinal epithelial mucosa by pathogens [40,41]. As a principal component of the chemical barrier, mucin secreted by goblet cells competitively binds bacteria at the binding sites of the intestinal epithelium, thereby enabling the bacteria to remain in the mucus layer and to be removed during intestinal motility [42]. To date, 21 mucin genes have been identified, most encoding secretory and membrane-bound mucins [43]. The progression of IBD is accompanied by a reduction in cupped cells and a decrease in mucin expression [19,23]. The findings indicate that LA mitigates DSS-induced intestinal epithelial cell damage and restores intestinal barrier function by reducing the expression levels of TNF-α, IL-1β, IL-6, and MDA, thereby attenuating inflammation and oxidative stress and upregulating ZO-1 and Occludin. Additionally, LA reverses the DSS-induced downregulation of mucins and intestinal barrier integrity.

Dysbiosis of the intestinal flora is linked to the etiology of several diseases [44]. In light of the intricate etiology of IBD, there is mounting recognition that intestinal flora dysbiosis plays a pivotal role in the pathogenesis of IBD. Previous reports have indicated that the composition of the intestinal flora of patients with IBD is altered compared to that of normal individuals [2,16,19]. This is evidenced by a decrease in the species diversity and abundance of the intestinal flora of patients with IBD and an increase in the abundance of several opportunistic pathogens. The β-diversity analysis revealed that the individuals in the control and DSS groups exhibited significant divergence from one another, whereas the DSS-LA group demonstrated a closer proximity to the control group. In addition, an increase in the abundance of *Proteobacteria*, including *Escherichia coli*, *Vibrio cholerae*, *Salmonella*, and numerous other pathogenic bacteria, was observed in the DSS group. This resulted in disruption to the microbiological composition of the mouse gut [45]. Furthermore, DSS treatment resulted in a notable increase in the prevalence of *Bacteroides*, a bacterium commonly associated with appendicitis and sepsis [46]. LA effectively reversed this alteration, enhancing the abundance of *Lactobacillus*, a beneficial microorganism for gut health. In general, LA enhances intestinal functionality by augmenting the quantity and diversity of the intestinal microbiota while restructuring its composition and elevating the proportion of beneficial bacteria compared to pathogenic bacteria.

## 5. Conclusions

This study isolated and screened a quality strain of *Lactobacillus acidophilus* LA from adult spotted hyena feces. It has a therapeutic effect on IBD by reducing inflammation and oxidative stress and upregulating the expression of TJ proteins, thereby restoring the intestinal barrier and remodeling the intestinal flora disorders caused by DSS. This is the inaugural investigation into the therapeutic effects of isolated probiotics derived from the scavenger spotted hyena on IBD. The findings offer promising insights into the potential of these probiotics as a source of novel therapeutic agents for IBD in the future.

## Figures and Tables

**Figure 1 nutrients-16-03682-f001:**
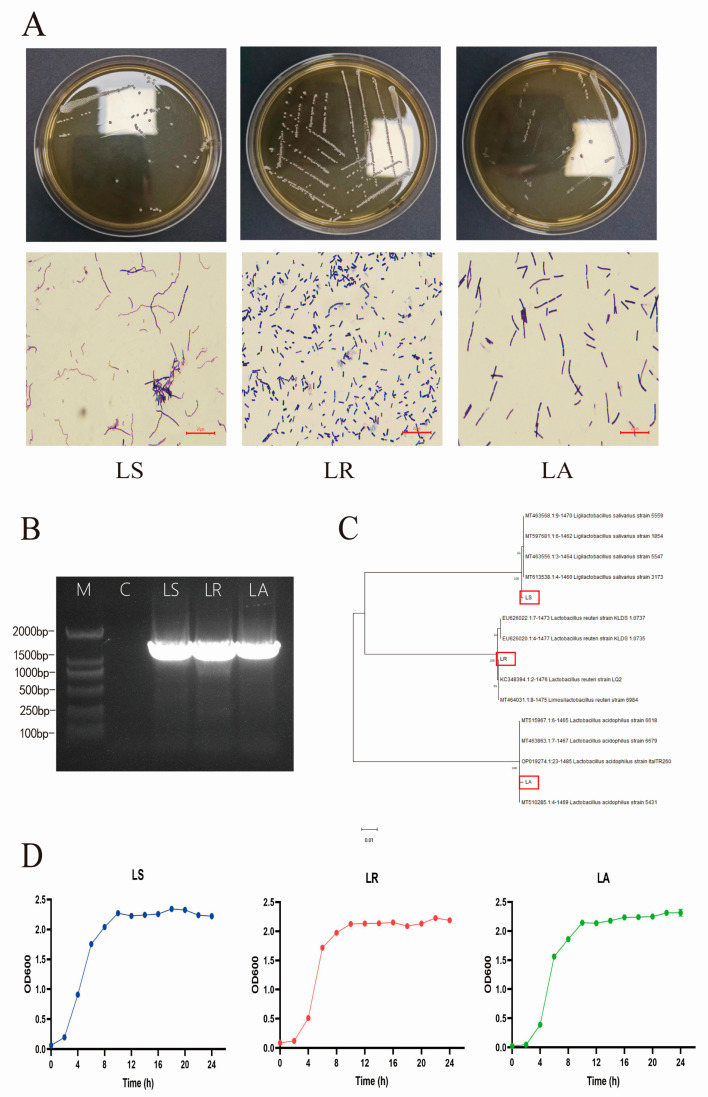
Isolation, identification, and growth characteristics of hyena-derived LAB. (**A**) The colony morphology and Gram staining diagrams of three isolates (100×). (**B**) The 16S rRNA agarose gel electrophoresis of the isolates’ PCR amplified product. M: 2000 bp DNA marker. (**C**) The phylogenetic tree using 16S rRNA gene sequencing of the isolates. The positions of the three isolates are indicated by red rectangles. (**D**) The growth curves of the isolates over 24 h. Scale bar: 20 μm. LS; *Lactobacillus salivarius*, LR; *Lactobacillus reuteri*, LA; *Lactobacillus acidophilus*.

**Figure 2 nutrients-16-03682-f002:**
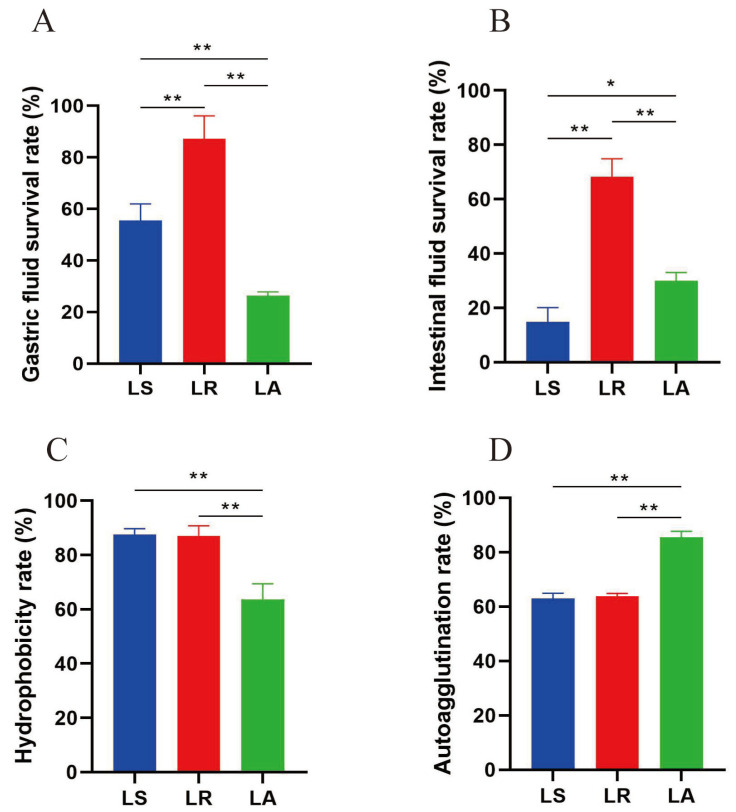
Survival and colonization of isolates in the gastrointestinal tract. (**A**) Gastric fluid survival rate. (**B**) Intestinal fluid survival rate. (**C**) Hydrophobicity rate. (**D**) Autoagglutination rate. * *p* < 0.05 and ** *p* < 0.01. LS; *Lactobacillus salivarius*, LR; *Lactobacillus reuteri*, LA; *Lactobacillus acidophilus*.

**Figure 3 nutrients-16-03682-f003:**
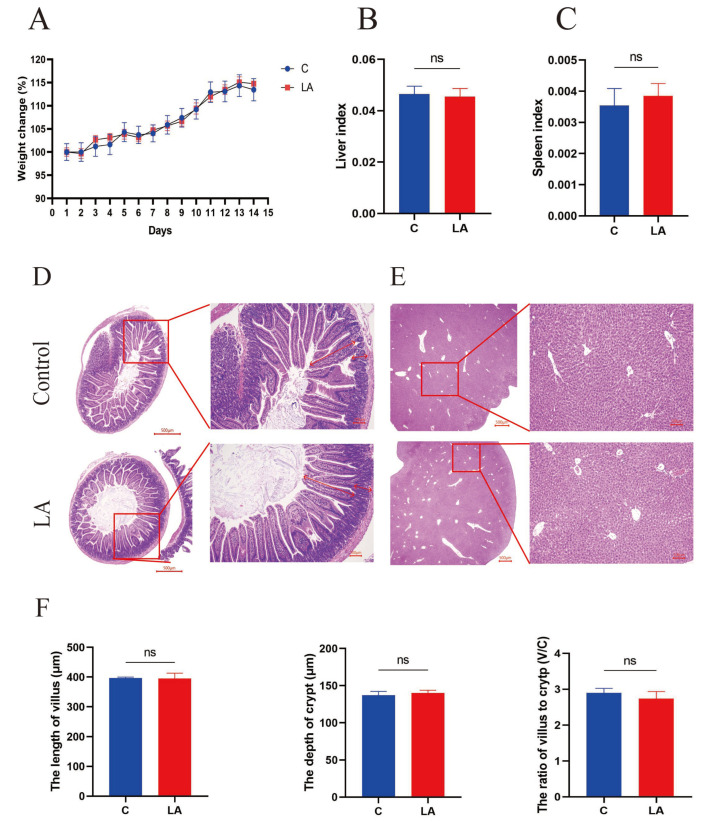
Safety assessment of isolated LA. (**A**) The weight change of mice in the control and LA groups. (**B**) The liver index of the mice. (**C**) The spleen index of the mice. (**D**) Histological sections of the jejunum (20× and 100×). (**E**) Histological sections of the liver. (**F**) The length of the villus and the depth of the crypt. Scale bar: 500 μm and 100 μm. ns *p* > 0.05. ns; not significant.

**Figure 4 nutrients-16-03682-f004:**
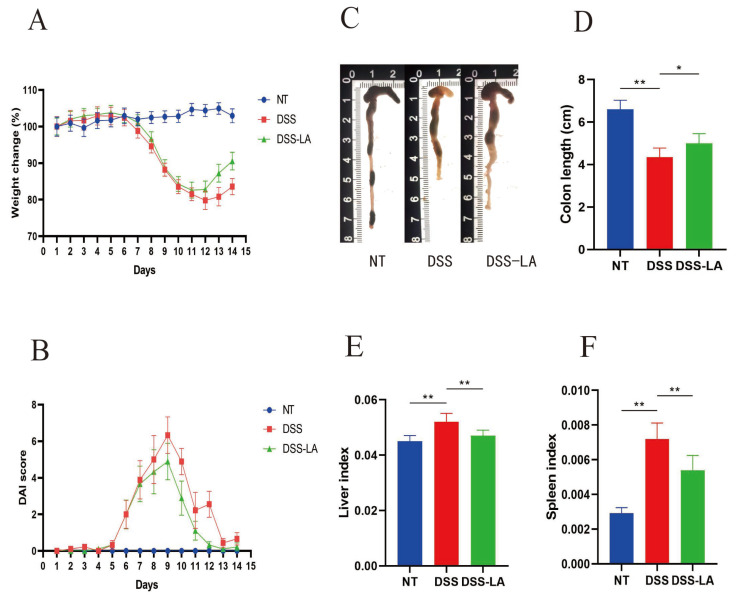
Effects of LA on DSS-induced symptoms. (**A**) Percent original body weight change following the induction of DSS-induced colitis. (**B**) Effects of LA on the disease activity index score of DSS-induced colitis. (**C**,**D**) Effects of LA on the colon length of mice after colitis induction. (**E**) The liver index of the mice. (**F**) The spleen index of the mice. * *p* < 0.05 and ** *p* < 0.01. NT; No Treatment.

**Figure 5 nutrients-16-03682-f005:**
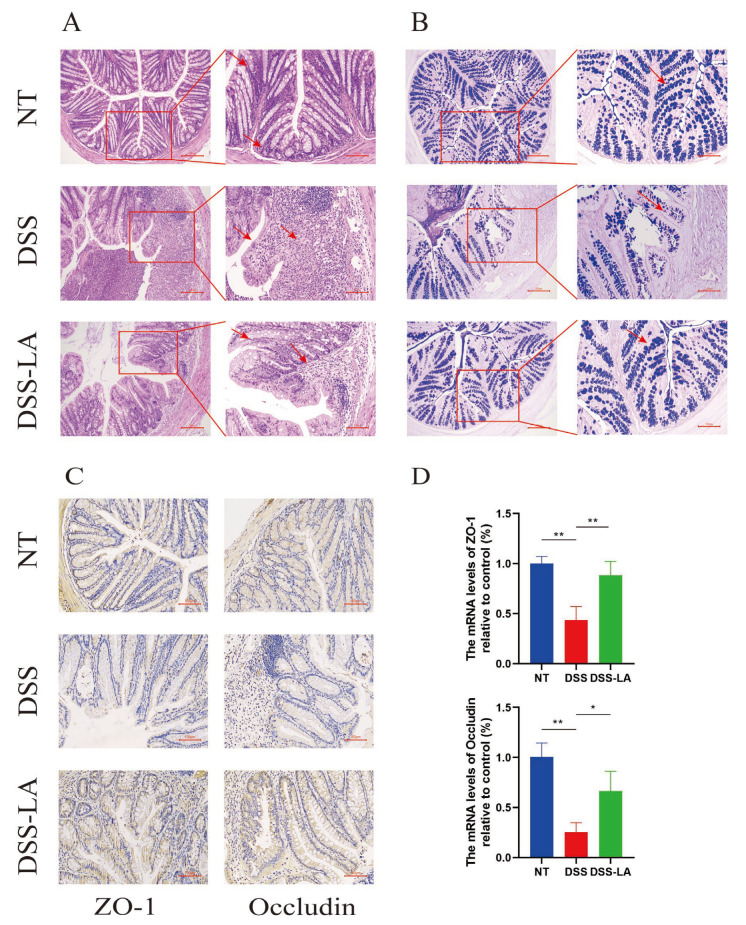
Effect of isolating LA on colon histology. (**A**) HE staining of colon tissue in different groups (100× and 200×). (**B**) Staining for Alcian blue-PAS carbohydrates showed significant loss of mucin (100× and 200×). (**C**) IHC to detect the expression level of TJ proteins (200×). (**D**) The expression of TJ proteins was determined via q-PCR. Scale bar: 200 μm and 100 μm. * *p* < 0.05 and ** *p* < 0.01.

**Figure 6 nutrients-16-03682-f006:**
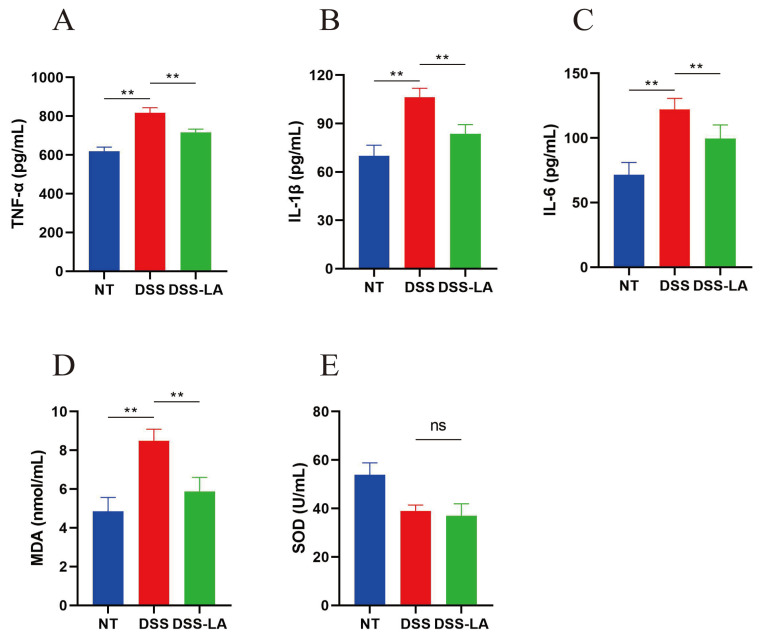
Effect of isolated LA on serum levels of inflammatory factors and oxidative stress in the mice. (**A**) TNF-α, (**B**) IL-1β, (**C**) IL-6, (**D**) MDA, and (**E**) SOD. ** *p* < 0.01.

**Figure 7 nutrients-16-03682-f007:**
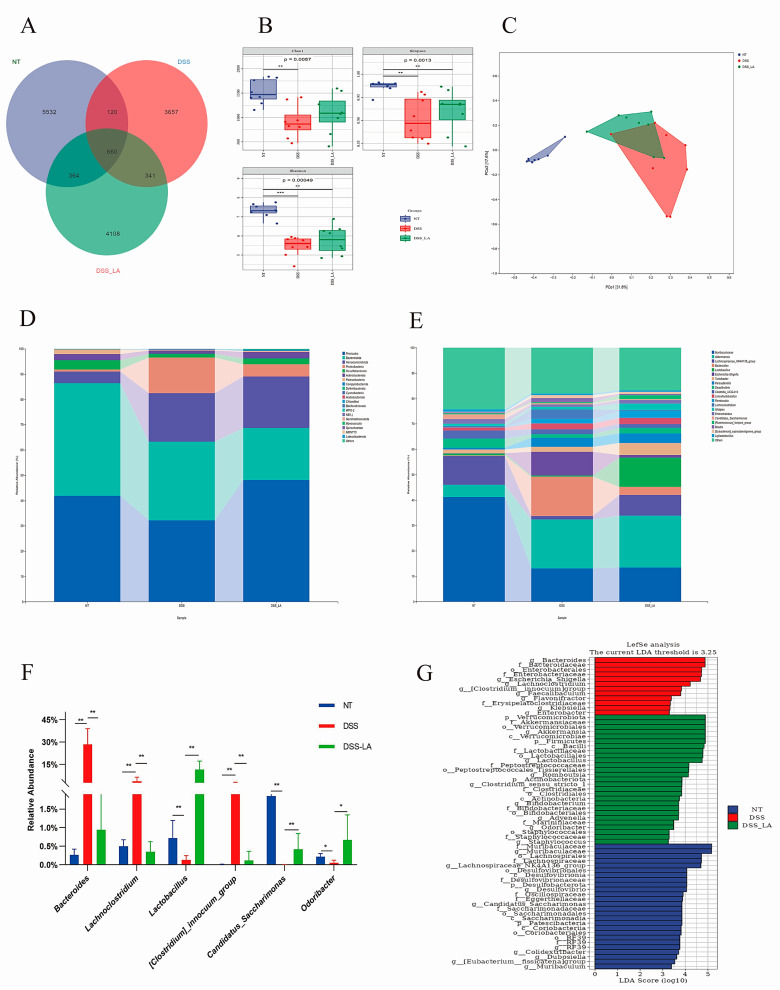
Effect of isolating LA on the intestinal flora of mice with DSS-induced colitis. (**A**) Venn diagram based on the samples’ OTUs. (**B**) Alpha diversity includes Chao1, Shannon, and Simpson indices. (**C**) Principal coordinates analysis (PCoA) of beta diversity. (**D**,**E**) The relative abundance of the gut microbial community at the phylum and genus levels. (**F**) Species with significant differences at the genus level. (**G**) Enriched taxa with a linear discriminant analysis score = 3.25 are shown in the histogram. Data are presented as means ± SD. * *p* < 0.05 and ** *p* < 0.01.

**Table 1 nutrients-16-03682-t001:** Oligonucleotide primers used for qRT-PCR.

Gene	Forward	Reverse
ZO-1	CTGGTGAAGTCTCGGAAAAATG	CATCTCTTGCTGCCAAACTATC
Occludin	TGCTTCATCGCTTCCTTAGTAA	GGGTTCACTCCCATTATGTACA
GAPDH	CCTTCATTGACCTTCACTACATGGTCTA	TGGAAGATGGTGATGGCCTTTCCATTG

**Table 2 nutrients-16-03682-t002:** Antibiotic drug sensitivity test of the isolates.

Antibiotic	LS	LR	LA
Amoxicillin	R	R	S
Spectinomycin	S	S	S
Florfenicol	R	R	S
Ceftriaxone	R	S	S
Cefalexin	R	R	S
Ampicillin	R	I	S
Cefuroxime	R	S	S
Minocycline	R	R	S
Chloramphenicol	I	S	S
Doxycyclinum	R	R	S

Notes: LS *Lactobacillus salivarius*, LR *Lactobacillus reuteri*, LA *Lactobacillus acidophilus*, S sensitive, R resistant, and I moderately sensitive.

## Data Availability

The datasets presented in this study are available from the NCBI BioProject number PRJNA 1149669.

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
