# Peer review of "The Therapeutic Effects of Lactic Acid Bacteria Isolated from Spotted Hyena on Dextran Sulfate Sodium-Induced Ulcerative Colitis in Mice"

_nutrients, 2024, doi:10.3390/nu16213682_

Round 1
Reviewer 1 Report
Comments and Suggestions for Authors
The manuscript by Xu et al is an interesting paper, with cognitive value and practical significance however, it requires a little tidying up. Both the introduction and methods are described properly and provide a good introduction to the discussed issue. The results are presented in a logical, orderly manner, however, some of the figures are illegible for the reader, difficult to interpret due to their small size (1C, 5A-B, 7 or histopathological images). Figure descriptions require clarification, no reference to type of applied statistics, no proper title of the figure (only explanations A), B)...fig.5 - ). The description should meet the condition of self-explanatory, with an explanation of the abbreviations used and general conditions of the study. Tables should also be self-explanatory. For example, Table 1 does not provide abbreviations LS, LR, LA. Moreover, Table 2 we have two times (no table 1). The numbering of tables and figures requires verification and correction. Discussion and conclusions are sufficient.
Reviewer 2 Report
Comments and Suggestions for Authors
Title:
Therapeutic effect of lactic acid bacteria isolated from spotted hyena on DSS-induced ulcerative colitis in mice
Xu M., Hu M., Han J., Wang L., He Y., Kulyar M.F., Zhang X., Lu Y., Mu S., Su H., Cao J., Li J.
Nutrients-3234062
In this manuscript, the authors tested the effect of three different Lactobacillus species, which they isolated from the feces of the scavenger-spotted hyena (Crocuta crocuta), as probiotics in DSS-induced IBD in mice. The results showed a significant improvement in the regulation of the intestinal flora as well as alleviating the symptoms, which is why Lactobacillus acidophilus LA1 is considered to have potential as a functional supplement.
Points of criticism:
At the end of the list of authors, there should probably be “and*” before the senior author.
Isolated typing errors and numerous blank space errors occur throughout the manuscript, especially when a reference number is given at the end and in the references to figures. These shortcomings must be corrected accordingly.
Lines 71 and 72 report that Proteles cristatus also feeds on termites, which is confirmed by reference 13. Citation 13, however, is about Parahyaena brunnea, with not a single mention of Aardwolf. In this regard, the authors should check this reference.
In line 94, a centrifugation step is skipped. The authors report the incubation on a shaker and start the next step with the supernatant. The centrifugation, including the centrifuge, time, and rpm, should be added.
Abbreviations must be explained the first time they are mentioned, e.g., MRS, NT or MDA.
The authors describe MDA as an antioxidant marker (lines 173-174) - in fact, it is an oxidative stress marker. In this regard, attention should also be paid to the title in chapter 2.9, as it refers to antioxidant enzymes (plural). In addition, the question arises as to whether there is any reference to these tests that could be cited. If this is not the case, a brief assay description is required.
In line 177, the authors refer to a methodology by Liu et al. - but without a reference, which needs to be added. Sequentially, it should be Ref 19, which shows Zhang Y. as the first author. This must be corrected.
In line 185, Table 2, referring to oligonucleotide primers, is mentioned. Since this is the first table in the manuscript and the sentence in line 183 refers to Table 1, this should be corrected accordingly.
In Table 2, chloramphenicol is probably meant instead of choramphenico.
The ordinate labelling in the middle figure of Figure 3F has been mixed up. Presumably, “depth of crypt” should be presented here.
The authors mention the “scale bar” in the tissue sections, although specifying the magnification would also be helpful.
The first sentence on page 11 (lines 300-301) and the last sentence (320-321) are obsolete and should be removed.
On page 14, line 357 (Liu et al.) and line 362 (Zhong et al.), the reference should be placed immediately after the name.
Reference No. 33 needs to be completed and must be supplemented accordingly.
